# Comparing COVID-19 Vaccination Outcomes with Parental Values, Beliefs, Attitudes, and Hesitancy Status, 2021–2022

**DOI:** 10.3390/vaccines10101632

**Published:** 2022-09-28

**Authors:** Tuhina Srivastava, Angela K. Shen, Safa Browne, Jeremy J. Michel, Andy S. L. Tan, Melanie L. Kornides

**Affiliations:** 1Department of Biostatistics, Epidemiology and Informatics, Perelman School of Medicine, University of Pennsylvania, Philadelphia, PA 19104, USA; 2Center for Public Health Initiatives, University of Pennsylvania, Philadelphia, PA 19104, USA; 3Leonard David Institute for Health Economics, University of Pennsylvania, Philadelphia, PA 19104, USA; 4Vaccine Education Center, Children’s Hospital of Philadelphia, Philadelphia, PA 19104, USA; 5Department of Medical Bioethics and Health Policy, Perelman School of Medicine, University of Pennsylvania, Philadelphia, PA 19104, USA; 6General Pediatrics, Children’s Hospital of Philadelphia, Philadelphia, PA 19104, USA; 7ECRI Guidelines Trust, ECRI, Plymouth Meeting, Philadelphia, PA 19462, USA; 8Annenberg School for Communication, University of Pennsylvania, Philadelphia, PA 19104, USA; 9Department of Family and Community Health, School of Nursing, University of Pennsylvania, Philadelphia, PA 19104, USA; 10Department of Pediatrics, Division of Adolescent Medicine, Perelman School of Medicine, University of Pennsylvania, Philadelphia, PA 19104, USA

**Keywords:** COVID-19 vaccines, vaccination hesitancy, parental decision making

## Abstract

Despite the availability of safe and effective COVID-19 vaccines, vaccine acceptance has been low, particularly among parents. More information is needed on parental decision-making. We conducted a prospective cohort study from October 2021 to March 2022 among 334 parents in a large urban/suburban pediatric primary care network and linked longitudinal survey responses about attitudes and beliefs on vaccination, social norms, and access to vaccination services for COVID-19 to electronic health-record-derived vaccination outcomes for their eldest age-eligible children in June 2022. The odds of accepting two doses of COVID-19 vaccine for their child was higher in respondents who indicated the COVID-19 vaccine would be very safe (aOR [CI]: 2.69 [1.47–4.99], *p* = 0.001), as well as those who previously vaccinated their child against influenza (aOR [CI]: 4.07 [2.08–8.12], *p* < 0.001). The odds of vaccinating their child were lower for respondents who attended suburban vs. urban practices (aOR [CI]: 0.38 [0.21–0.67], *p* = 0.001). Parents in the cohort were active users of social media; the majority (78%) used their phone to check social media platforms at least once per day. Our findings suggest that healthcare providers and policymakers can focus on improving vaccination coverage among children living in suburban neighborhoods through targeted mobile-based messaging emphasizing safety to their parents.

## 1. Introduction

COVID-19 vaccines have successfully reduced the morbidity and mortality associated with COVID-19 globally and have helped millions around the world avoid severe illness and hospitalization and return to daily activities [1,2]. Achieving high community immunity, largely through high vaccination coverage rates, is a key strategy to mitigating the devastating impacts of the COVID-19 pandemic [3]. However, vaccination coverage rates for the uptake of two doses of the COVID-19 vaccine fell short of the 80% coverage goal to achieve community immunity in the United States (U.S) for the following groups as of 10 August 2022: 5–11 years (30.2%), 12–17 years (60.2%), 18–24 years (64.7%), and 25–49 years (70.4%) [3,4]. While vaccines for children under age five years have recently received emergency use authorization (EUA), only 3.1% of children <2 years and 5.2% of children 2–4 years of age have initiated vaccination between 18 June 2022 and 15 August 2022 [4]. New strategies to increase vaccination coverage among children have been launched; for example, in December 2021 the Centers for Medicare and Medicaid Services (CMS) implemented 100% federal financing for COVID-19 vaccine counseling visits, inclusive of routine vaccines, for children and youth insured through the Medicaid program, a vulnerable population of children from low-income families, many of whom are Black and Latinx [5].

While most pediatric infections are asymptomatic or clinically mild, there have been cases of pediatric hospitalization and mortality from COVID-19 or related complications [6]. Almost 8800 children have experienced multisystem inflammatory syndrome in children (MIS-C). Many children have struggled with the mental health consequences of online schooling, social isolation, disruption in daily activities, and grief resulting from the loss of loved ones [7,8,9,10]. Low-income and racial or ethnic minority children have been disproportionately impacted [11]. Many parents have difficulty in making the decision to vaccinate their children because cases of COVID-19 are perceived as asymptomatic or mild in children, and so they have taken a “wait-and-see” position when it comes to vaccinating their children [12]. Previous studies on parental perceptions on COVID-19 vaccines and intent to vaccination document a clear need for more information about the vaccines, particularly from health care providers, a trusted source on vaccines [12,13,14]. Concerns about COVID-19 vaccines echo sentiments about other childhood vaccines, and vaccine hesitancy has been a growing concern worldwide [15,16,17,18,19]. In the U.S., approximately 6% of parents have some form of hesitancy around routine vaccines. About 26% are hesitant to give their children influenza vaccine, and about 23% resist human papillomavirus vaccine for their children [20,21,22]. Parents’ concerns are anchored around vaccine safety and effectiveness relative to risk.

Our objective was (1) to understand parental attitudes and beliefs on vaccination, social norms, and access to vaccination services for COVID-19 and (2) characterize associations between parental beliefs and EHR-documented vaccination uptake as the COVID-19 vaccine became available to children aged 12 years and older, and subsequently to children aged 5 years and older in the United States.

## 2. Materials and Methods

We conducted a prospective cohort study in which we deployed two longitudinal surveys and linked them to COVID-19 vaccination outcomes in the Children’s Hospital of Philadelphia (CHOP) Outpatient Care Network. The network included 31 pediatric primary care sites located within the greater Philadelphia regions: 3 academic and 3 non-academic sites located within Philadelphia, and 25 suburban sites located across the Philadelphia (*n* = 22) and New Jersey (*n* = 3) suburbs (Appendix A). Academic sites have a higher proportion of uninsured and Medicaid patients and include physicians-in-training [23].

Using the CHOP electronic health record (EHR) (Epic Systems, Inc., Verona, WI, USA) we recruited parents and legal guardians (referred to collectively as “parents” in this article) of current pediatric patients through 19 years of age seen for a preventative care visit at one of the outpatient general pediatric practices from 1 January 2019 to 31 August 2021. To arrive at the final list of 40,583 parents invited to participate, we de-duplicated parents with multiple children seen at CHOP, those who opted out of research, and those whose children were deceased. Two sequential surveys were administered using Research Electronic Data Capture (REDCap) (Figure 1). Those who completed informed consent and survey one, conducted from 12 October 2021 to 21 January 2022, received the follow-up survey between 27 January and 18 February 2022 (Figure 1). Data extracted from the EHR include name, age, race, insurance status, recent visit timing, immunization records, ethnicity, birth gender, primary contact email address, and all associated parents on the chart through 18 June 2022. The CHOP Institutional Review Board approved the study (IRB 21-019115).

*Survey Instruments*. Survey items included questions on perceptions of routine immunization, potential hesitancy, attitudes, and confidence about COVID-19 vaccines. Other items explored how to address barriers to access and communicate with parents about information they would like to receive on the COVID-19 vaccines (Appendix A). In addition to collecting demographic information, we asked parents about vaccine hesitancy and explored domains on feelings and motivation on social process and return to school for children, as well as access to vaccination services (Question 21, Appendix A), In the follow-up survey (Survey 2), we explored any changes in parental attitudes, beliefs, and confidence in the interim (Appendix A). Survey 1 was administered after the vaccine was approved for use in children ages 12 and older in the U.S., and Survey 2 was administered after the vaccine was approved for use in children ages 5–11 years.

*Vaccine Attitudes.* We used the following existing, validated scales and/or questions to measure various attitudes and beliefs:Modified Vaccine Hesitancy Scale (VHS): A 9-item scale (with responses on a 4-point Likert-type scale ranging from Strongly agree, Agree, Disagree, to Strongly disagree) adapted from the Vaccine Hesitancy Scale for childhood vaccines [20].Kaiser Family Foundation (KFF) COVID-19 Vaccine Monitor: This ongoing research project uses both surveys and qualitative research to understand, in part, parental attitudes and experiences with COVID-19. We used and adapted reported questions from this resource (Questions 23, 29, 35–36 in Appendix A and Questions 32–33 in Appendix A) [12].CDC Vaccine Confidence Survey (Questions 38–40 in Appendix A and Questions 36–38 in Appendix A) [24].Vaccination Attitudes Examination (VAX) Scale: A 12-item scale (with responses on a 6-point Likert-type scale ranging from (1) “strongly agree” to (6) “strongly disagree”) created to understand the following: trust/mistrust of vaccine benefit, worries over unforeseen future effects, concerns about commercial profiteering, and preference for natural immunity (Questions 6–17 in Appendix A) [25]. The first three items were reverse-coded so lower total scores reflected stronger anti-vaccination attitudes, and we summed all twelve item scores to obtain a composite score as well as a median score. The four themes above corresponded to validated subscales, which we calculated by summing the scores from the three items in each subscale.


Other Scales Used:Coronavirus Anxiety Scale: A 5-item screening tool to identify probable cases of dysfunctional anxiety associated with the coronavirus (Questions 41–45 in Appendix A) [26].Media and Technology Usage and Attitudes Scale (MTUAS): The overall MTUAS was created to measure media and technology involvement of respondents [27]. We used the 9-item General Social Media Usage Subscale (Questions 46–54 in Appendix A).


Statistical Analysis.

Summary statistics and bivariate associations with the outcome (below) were performed.

*Outcome.* Using EHR data, we determined whether each age-eligible child of participating parents had received two doses of a COVID-19 vaccine. We characterized this into a binary COVID-19 primary series completion status variable (Yes/No) for the eldest age-eligible child of each respondent.

*Logistic Regression.* We modeled differences between parents whose age-eligible eldest child had versus had not received both doses of COVID-19 vaccine using a multivariable logistic regression model (α = 0.05) and have reported adjusted odds ratios with 95% confidence intervals and *p*-values. Variable selection for our final logistic regression was performed using Lasso regression (α = 1) [28]. All selected variables were checked for missingness (using a 5% maximum missing threshold) and collinearity (using a variance inflation factor threshold of less than 10). We performed sensitivity analyses by additionally running separate models for children ages 5–11 years and 12–15 years at the date of Survey 1 completion, based on the age groups approved by the FDA (Figure 1).

All analyses were performed using RStudio 2022.07.1 running R 4.1.3.

## 3. Results

Of 40,583 total invitations, 1259 participants completed Survey 1 (3.1% response rate), and 538 of those participants completed Survey 2 (43% study completion rate for Survey 2 from Survey 1). From those, a total of 334 respondents had age-eligible children for COVID-19 vaccination during our follow-up time and included in the current study. Respondents’ median age was 41 years [95% confidence interval (CI): 24–64 years], most identified as female (*n* = 307, 91.9%), and most identified as heterosexual/straight (*n* = 300, 89.8%) (Table 1). Most (*n* = 238, 71.3%) respondents identified as “White or Caucasian”, and the majority (*n* = 319, 95.5%) stated not having any Hispanic, Spanish, or Latin origin. Most of the respondents (*n* = 380, 71%) indicated earning over USD 100,000/year, and most parents had attained some college degree (*n* = 268, 80.2%) (Table 1). Most respondents had already received two doses of a vaccine (*n* = 310, 92.8%), and had also received an influenza vaccine in the past year (*n* = 256, 76.6%) (Table 1). Almost all of their eldest children had received all recommended vaccines for their age (*n* = 328, 98.2%). Only 23 (6.9%) respondents could be characterized as vaccine hesitant, and about 70% (*n* = 233) of parents with age-eligible children had vaccinated their eldest child against COVID-19 with two vaccine doses (Table 1) [20]. The majority of respondents indicated that they knew someone who had been infected with COVID-19, with 29.3% (*n* = 98) answering “Yes, me”, and 56.9% (*n* = 190) answering “Yes, a close family member or friend.”

### 3.1. Vaccine Access, Beliefs, and Attitudes

#### 3.1.1. Vaccine Access

When asked to select their top choice, the majority (*n* = 204, 61.1%) of respondents stated they would prefer to have their child vaccinated at their pediatrician’s or physician’s office, but most, (*n* = 74, 22.2%), indicated they had no preference on location.

#### 3.1.2. Vaccine Reasons and Hesitancy

Using the modified Vaccine Hesitancy Scale by Helmkamp, et.al., we found that most respondents were not hesitant (*n* = 311, 93.1%) as compared to hesitant (*n* = 23, 6.9%). When asked to select the statement they agreed with most in response to “getting my child vaccinated will…”, respondents answered: protect my child (*n* = 60, 18%), protect family/friends (*n* = 9, 2.7%), protect others in school/community (*n* = 6, 1.8%), allow my child(ren) to resume travel (*n* = 4, 1.2%), allow my child(ren) to resume social activities and sports (*n* = 2, 0.6%), and all of the above (*n* = 239, 71.6%).

However, some respondents criticized COVID-19 vaccines, objecting to vaccination for their children. One said they did not want to “put toxins in [their child’s] body”, another said that they are “against giving my child c-19 vaccines—it gave me long-lasting side effects,” and another one claiming vaccines to “be unnecessary for an everchanging virus and underaged vaccines created for financial gain.”

#### 3.1.3. Vaccine Attitudes Examination (VAX) Scale

The overall median score for all 12 items of the VAX scale was 5 [1,2,3,4,5,6], indicating generally positive overall attitudes about vaccination (Figure 2). The full VAX scale included the following subscales: trust/mistrust of vaccine benefit (5 [1,2,3,4,5,6]), worries over unforeseen future effects (4 [1,2,3,4,5,6]), concerns about commercial profiteering (5 [1,2,3,4,5,6]), and preference for natural immunity (5 [1,2,3,4,5,6]).

### 3.2. Coronavirus Anxiety Scale

Most respondents answered “Not at all” on a Likert scale when asked how often over the last 2 weeks they had experienced anxiety-related symptoms when thinking about or seeing information about coronavirus. Only 8% indicated feeling dizzy, lightheaded, or faint, and only 9% reported any loss of appetite. About 14% indicated experiencing nausea or stomach problems, and 11.7% reported feeling paralyzed or frozen. The most frequently reported effect was trouble falling or staying asleep (24.6%).

### 3.3. Non-Pharmaceutical Interventions and Behaviors

When asked about various social behaviors in which they would participate in with their child if they themselves were vaccinated, most parents responded that they were “Somewhat likely” “or “Very likely” to wear a mask while indoors (*n* = 295, 88.3%), stand 6 feet away from others (*n* = 290, 86.8%), avoid public transit (*n* = 238, 71.3%), avoid indoor dining (*n* = 208, 62.3%), and avoid indoor gatherings (*n* = 224, 67.1%). Over half (*n* = 178, 53.3%) of respondents stated that they were “Not at all likely” to avoid traveling out of state.

### 3.4. School Restrictions and Behaviors

The majority of parents answered “Yes” when asked if their children’s school should implement the following safety measures: require students to be vaccinated for COVID-19 (once FDA authorized for all school-age children) as they do for most other diseases such as measles (*n* = 241, 72.2%); require unvaccinated students and staff to wear masks (*n* = 292, 87.4%); provide voluntary, free weekly COVID-19 testing of children at schools (*n* = 274, 82%).

### 3.5. Social Media Usage

Most participants have Facebook (*n* = 257, 76.9%) and Instagram (*n* = 229, 68.6%) accounts, as well as Pinterest (*n* = 114, 34.1%), Twitter (*n* = 117, 35%), TikTok (*n* = 74, 22.2%). When asked about their general social media usage using the Media and Technology Usage and Attitudes Scale (MTUAS), about half (*n* = 153, 46%) of respondents said they “never” check their social media pages from their computer but rather the majority (*n* = 261, 78%) said they check their social media from their phone at least once a day or more frequently (Figure 3). Over half of respondents said they posted updates (*n* = 180, 54%) or photos (*n* = 190, 57%) once a month or even less frequently, yet most respondents said they would browse profiles and photos (*n* = 202, 60%), read postings and/or watch videos (*n* = 253, 76%), and like a posting, video, update, photo, etc. (*n* = 215, 64%) more than once a week. Commenting on postings, videos, updates, photos, etc. was less frequent with only 46% (*n* = 152) of respondents saying they did this more than once a week.

### 3.6. Multivariable Logistic Regression

In the multivariable model, respondents who indicated the COVID-19 vaccine would be very safe had much higher odds of vaccinating their eldest child against COVID-19 (aOR [CI]: 2.69 [1.47–4.99], *p* = 0.001). The odds of accepting two doses of COVID-19 vaccine for their eldest child was about two times higher (aOR [CI]: 2.12 [1.04–4.36], *p* = 0.039) in respondents with a college degree compared to those with no college degree. Having previously vaccinated their eldest child against influenza was also associated with four times higher (aOR [CI]: 4.07 [2.08–8.12], *p* < 0.001) odds of already vaccinating against COVID-19 (Table 2) compared to those who had not. The odds of vaccinating their eldest child were 2.6 times lower (aOR [CI]: 0.38 [0.21–0.67], *p* = 0.001) for respondents who took their children to suburban practices compared to Philadelphia practices (Table 2). Sensitivity analyses were performed for the 5–11- and 12–15-year age cohorts; however, low observation numbers yielded unstable estimates.

## 4. Discussion

In this longitudinal study of two waves of parental surveys over a five-month period linked to vaccination outcomes in a pediatric primary care network in the greater Philadelphia area, we assessed parental values, beliefs, attitudes, and vaccine hesitancy status and measured associations between COVID-19 vaccine acceptance for their eldest child and myriad variables. Most parents in this cohort were not vaccine-hesitant, ensuring they had received an annual influenza vaccine for themselves and their children, and their children were up-to-date with routinely recommended vaccinations. However, even among these parents who generally accepted vaccines, the COVID-19 vaccine was not universally accepted for their eldest children. We found one of the most important predictors of COVID-19 vaccine acceptance was parents’ belief in the safety of the vaccine. This is consistent with previous studies which have linked parents’ anxiety over COVID-19 vaccines side effects and safety with vaccination refusal for their children [29,30]. Healthcare providers, who remain a main source of trusted information for parents, should continue to recommend vaccination as a provider recommendation increases the likelihood that parents will vaccinate their children [29,31,32]. Providers and public health practitioners can develop robust talking points to address parental questions about the safety of these vaccines when counseling parents of young children, as well as provide easy, frequent access to these vaccines, such as at their physician’s office, in order to boost vaccine uptake, especially in low-coverage areas [29].

Parents most often cite concerns about vaccine safety and in the case of COVID-19 long-term safety consequences. Given the emergency circumstances under which COVID-19 vaccines were authorized, especially in the context of changing community case rates and shifts in school and public policy around masking and social distancing, parents have struggled with how to best support their children [12,18]. Most of the parents in this study supported preventive behaviors such as such as continuing to mask while indoors, standing 6 feet away from others, avoiding public transit, avoiding indoor dining, and avoiding indoor gatherings. They also believed that unvaccinated students and staff should remain masked in schools, suggesting that they favor more precautionary behavior in order to protect their children. Furthermore, regardless of their COVID-19 vaccine decision for their children and despite many delaying their child’s vaccination, over 90% of parents still believed that COVID-19 vaccination would protect their child. These data suggest that parents want to have agency in protecting their children against COVID-19 through preventive behaviors, ultimately choosing “layers” of protection with masking, distancing, vaccinating, and other. They also care about the safety and side effects of vaccines as they think about vaccinating their children. The “newness of the vaccine” has been difficult for individuals to grapple with given the imperative to vaccinate as soon as vaccine supply was available [9,12]. Given that parents continue to trust healthcare providers for reliable information, continuing conversations with them about COVID-19 vaccine safety and any other issues parents may have can increase confidence about vaccination and may increase uptake among parents continue to delay vaccination [9]. With newly reformulated boosters expected to come out in this fall, it will be important to ensure easy access to trusted providers for parents to continue to have conversations about initiating vaccination for those unvaccinated, and completing or boosting for those who have started the vaccine series [33].

Parents also report being to an overload of information, changing over time, which results in exposure to both credible information as well as misinformation about vaccines [18,34]. Social media is saturated with vaccine information, yet exposing parents to accurate, useful information repeatedly could help increase vaccine confidence [9,18,34]. Responses to questions about social media usage indicated that most parents primarily check their social media sites on their phone and passively consume content by browsing and watching rather than engaging by commenting on posts. Additionally, parents continually emphasize that physicians are a trusted source of information and having respectful conversations with them empowered them to make informed vaccine decisions [9]. Having an understanding of how and where parents are obtaining information (e.g., social media, cell phones, etc.) can help ensure that information is reaching them where they are looking in order to promote COVID-19 vaccine uptake. Creating and disseminating phone-friendly content from healthcare organizations could strategically be used to target messaging on Facebook and Instagram to reach more parents still deciding about vaccination.

Regarding measuring parental hesitancy for practitioners and researchers, while most parents indicated largely positive attitudes on the VAX scale, this validated measure of parental attitudes towards routine vaccines was not a significant predictor of vaccination in our study and was not included in our final model. This could suggest that existing vaccine hesitancy scales developed before COVID-19 are not a good proxy for COVID vaccine uptake, particularly not in a cohort in which most parents do not report vaccine hesitancy. This also concurs with our focus groups from this cohort which showed that vaccine decision making is complex and stems from myriad factors such as individual influences, group influences, vaccine and vaccine program influences, and contextual influences [9]. Interestingly, in the current study, we found previous uptake of recommended vaccines, such as an influenza vaccine, to be an important predictor of COVID-19 vaccine uptake. This suggests that previous vaccination behavior may be a more accurate predictor of vaccine hesitancy around the COVID-19 vaccine decision making.

The large disparity in the odds of COVID-19 vaccine uptake in suburban parents compared to urban parents could be indicative of different information exposure or access and different risk assessment, which could affect decision making related to COVID-19 vaccination specifically [35]. Parents taking their kids to suburban practices most likely also live in the suburbs of Philadelphia, and may represent differences in political ideology, as the suburbs of Philadelphia are slightly more conservative than Philadelphia County [36]. Prior studies have found that parental hesitancy towards COVID-19 vaccination has varied by political ideology [18]. Additionally, prior studies have found that suburban parents have lagged in acceptance of vaccines compared to those living in urban areas, with a larger share of suburban parents reporting lower confidence in the safety of COVID-19 vaccines, even when controlling for political ideology [35]. As parents continue to cite pediatricians as their top trusted source of vaccine information for their children across community types including suburban vs. urban, our finding highlights the importance of healthcare providers to initiate repeated, respectful conversations for parental concerns, especially regarding vaccine safety among suburban residing parents [35]. This is particularly important as new federal funding has been allocated for COVID-19 and routine vaccine counseling visits; therefore providers, especially those in low-coverage areas such as the suburbs, need to use these opportunities to offer clear explanations to address parental concerns [37].

### Limitations

Limitations of this study include selection bias in respondents given our convenience sample, missingness in responses to non-mandatory questions, and logistical constraints with timing of survey administration. First, there may be selection bias in those who responded to the survey. Since there was a low response rate to Survey 1, parents who responded may be those who did not have technological constraints (biasing results to more affluent families) or felt strongly about or positively towards vaccines. Additionally, many parents were healthcare professionals who may be more inclined to complete both surveys and also more inclined to vaccinate their children. We hypothesize these parents were more likely to vaccinate due to their (1) knowledge and experience with COVID-19 diseases and (2) potential concerns regarding their own increased exposure to COVID-19 in the workplace. Additionally, this group of parents was largely Non-Hispanic White, female, and vaccine acceptors, resulting in limited breadth of perspectives. Next, data for many questions had over 50% missingness and had to be excluded from our analysis because not all survey questions were made mandatory for survey completion. Another limitation is the timing of the survey administration—given logistical computing constraints for the volume of initial invites, the first survey was be sent out over the span of multiple months, and age-related recommendations changed during the survey timeframe; therefore, causality of governmental recommendations on vaccine behaviors cannot be established from this cohort. Additionally, by using COVID-19 vaccine outcomes from only the eldest child we limited our data set; however, this ensured that we did not oversample parents with multiple children. Finally, these analyses are limited to participants from one geographic area in the mid-Atlantic region. While our results cannot be generalized to the overall U.S. population, our sample is diverse and represents both urban and suburban populations. Given this large, longitudinal cohort, this paper provides compelling, initial insight on parental considerations and decision making about vaccinating their eligible children against COVID-19 by linking survey responses to vaccine uptake outcomes and identifies populations to target for future interventions.

## 5. Conclusions

COVID-19 vaccine decision making may differ from routine vaccinations given the perceived newness of COVID-19 vaccines, and vaccine safety remains at the core of parental concerns, despite their vaccine behaviors for other routine vaccines. The pandemic circumstances augmented these concerns through emergency use authorization pathway and the onslaught of information surrounding them. Healthcare providers and policymakers should seek to understand specific parental concerns, focusing on parents living in suburban or other low-coverage neighborhoods, and improve vaccination coverage through targeted mobile-based messaging, creating safe spaces to discuss vaccine concerns, and providing convenient opportunities to vaccinate.

## Figures and Tables

**Figure 1 vaccines-10-01632-f001:**
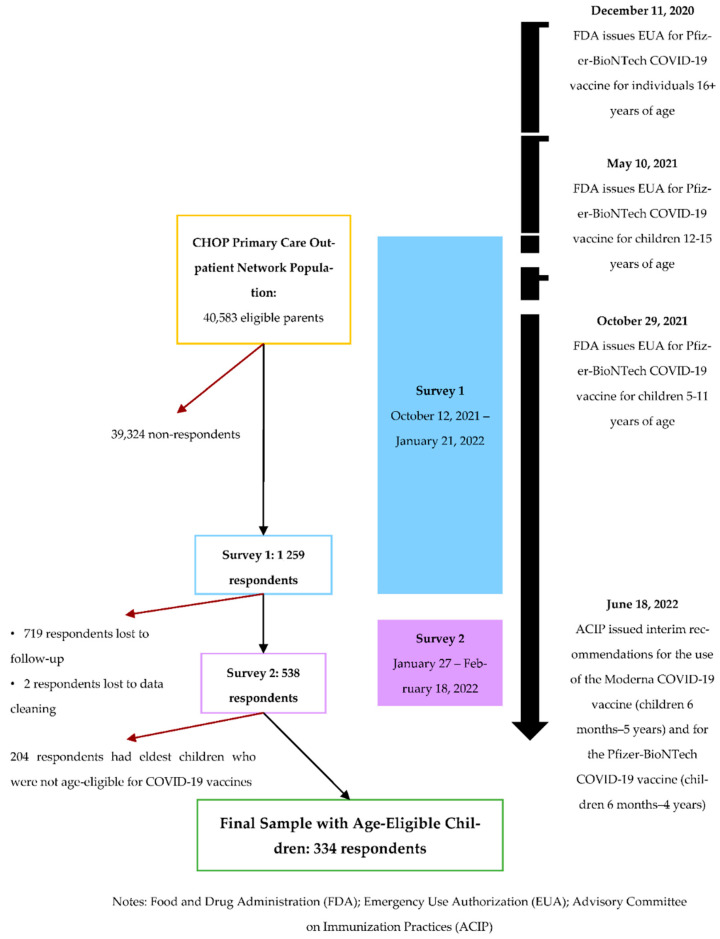
Study Recruitment for Longitudinal COVID-19 Vaccine Surveys from Children’s Hospital of Philadelphia (CHOP) Outpatient Primary Care Network, 2021–2022.

**Figure 2 vaccines-10-01632-f002:**
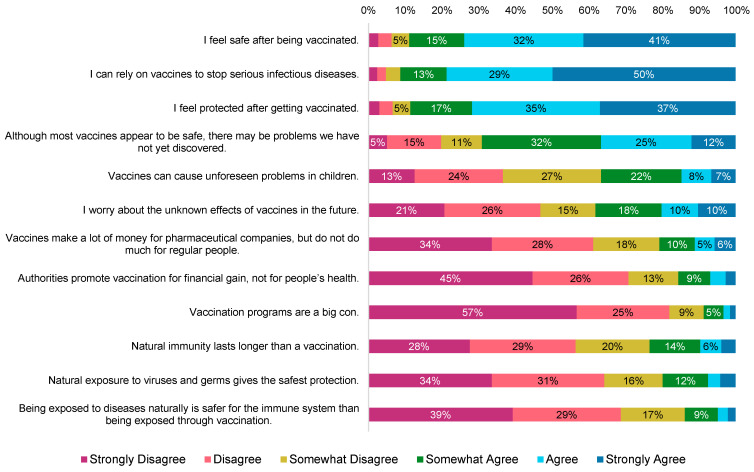
Responses to the Vaccination Attitudes Examination (VAX) Scale. Note: Values are represented as a percentage of the overall total (*n* = 334), and data labels for response values lower than 5% have been excluded from this figure. Given the reverse directionality of responses for the first three items, they were reverse-coded when calculating the overall VAX scale score, but show the original responses here.

**Figure 3 vaccines-10-01632-f003:**
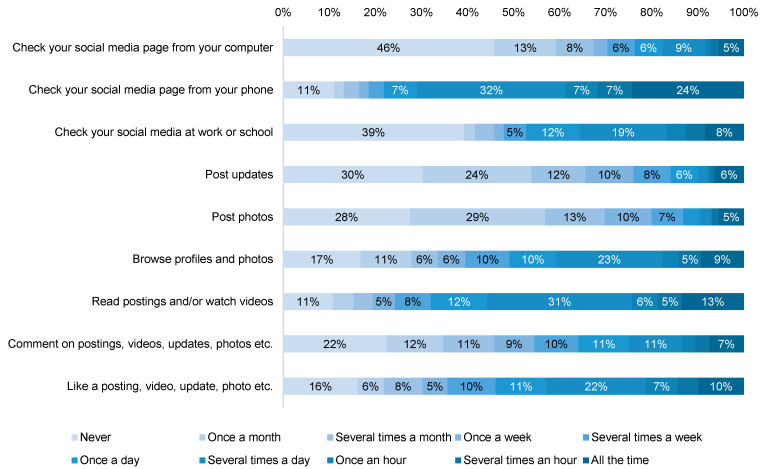
Responses to the Media and Technology Usage and Attitudes Scale (MTUAS) General Social Media Usage Subscale. Note: Values are represented as a percentage of the overall total (*n* = 334), and data labels for response values lower than 5% have been excluded from this figure.

**Table 1 vaccines-10-01632-t001:** Sociodemographic characteristics of survey respondents with age-eligible children, stratified by receipt status of two doses of COVID-19 vaccine for their eldest child (*n* = 334).

Parental Characteristics	No, Is Age-Eligible But Did Not Receive Two Doses of COVID-19 Vaccine	Yes, Received Two Doses of COVID-19 Vaccine	Total	Χ^2^ *p*-Value
	(*n* = 101)	(*n* = 233)	(*n* = 334)	
**Parental Age**				0.048 *
Under 35 years	23 (22.8%)	28 (12.0%)	51 (15.3%)	
35–38 years	21 (20.8%)	44 (18.9%)	65 (19.5%)	
38–43 years	21 (20.8%)	71 (30.5%)	92 (27.5%)	
Over 43 years	36 (35.6%)	88 (37.8%)	124 (37.1%)	
**Sex (assigned at birth)**				0.745
Male	7 (6.9%)	19 (8.2%)	26 (7.8%)	
Female	94 (93.1%)	213 (91.4%)	307 (91.9%)	
**Gender Identity**				0.111
Man	7 (6.9%)	19 (8.2%)	26 (7.8%)	
Woman	90 (89.1%)	211 (90.6%)	301 (90.1%)	
Nonbinary	2 (2.0%)	1 (0.4%)	3 (0.9%)	
Prefer not to answer	0 (0%)	2 (0.9%)	2 (0.6%)	
Other	2 (2.0%)	0 (0%)	2 (0.6%)	
**Sexual Orientation**				0.548
Heterosexual or straight	91 (90.1%)	209 (89.7%)	300 (89.8%)	
Gay or lesbian	3 (3.0%)	3 (1.3%)	6 (1.8%)	
Bisexual	3 (3.0%)	9 (3.9%)	12 (3.6%)	
Prefer not to answer	2 (2.0%)	10 (4.3%)	12 (3.6%)	
Different identity	2 (2.0%)	2 (0.9%)	4 (1.2%)	
**Race**				0.962
White or Caucasian	71 (70.3%)	167 (71.7%)	238 (71.3%)	
Black or African American	19 (18.8%)	41 (17.6%)	60 (18.0%)	
Other	11 (10.9%)	25 (10.7%)	36 (10.8%)	
**Hispanic Origin**				0.790
Not Hispanic	96 (95.0%)	223 (95.7%)	319 (95.5%)	
Hispanic	5 (5.0%)	10 (4.3%)	15 (4.5%)	
**Income**				0.936
Under USD 150,000/year	59 (58.4%)	135 (57.9%)	194 (58.1%)	
Over USD 150,000/year	42 (41.6%)	98 (42.1%)	140 (41.9%)	
**Education**				0.035 *
No college degree (High school diploma, GED, some college credit)	27 (26.7%)	39 (16.7%)	66 (19.8%)	
College degree	74 (73.3%)	194 (83.3%)	268 (80.2%)	
**Insurance Type**				0.265
Public	32 (31.7%)	60 (25.8%)	92 (27.5%)	
Private	69 (68.3%)	173 (74.2%)	242 (72.5%)	
**Practice Type**				0.011 *
Philadelphia	46 (45.5%)	141 (60.5%)	187 (56.0%)	
Suburban	55 (54.5%)	92 (39.5%)	147 (44.0%)	
**Number of Children**				0.968 ^#^
Median [Min, Max]	2 [1, 4]	2 [1, 5]	2 [1, 5]	
**Child Age, Eldest Child**				0.068 ^#^
Median [Min, Max]	9 [1, 18]	10 [1, 18]	10 [1, 18]	
**COVID-19 Vaccination Status, Parent**				<0.001 *
Yes	79 (78.2%)	231 (99.1%)	310 (92.8%)	
No	22 (21.8%)	2 (0.9%)	24 (7.2%)	
**First Dose of COVID-19 Vaccine Received, Eldest Child**				<0.001 *
Yes, received	12 (11.9%)	233 (100%)	245 (73.4%)	
No, but is eligible	89 (88.1%)	-	89 (26.6%)	
**Influenza Vaccination Status, Parent**				<0.001 *
Yes	66 (65.3%)	190 (81.5%)	256 (76.6%)	
No	35 (34.7%)	42 (18.0%)	77 (23.1%)	
**Influenza Vaccination Status, Eldest Child**				<0.001 *
Yes	68 (67.3%)	209 (89.7%)	277 (82.9%)	
No	33 (32.7%)	24 (10.3%)	57 (17.1%)	

Χ^2^ = Chi-squared. * denotes significant *p*-value (α = 0.05). ^#^ Mann–Whitney U test or Wilcoxon Rank-Sum test.

**Table 2 vaccines-10-01632-t002:** Multivariable logistic regression results for two-dose COVID-19 vaccine receipt status of age-eligible eldest children of survey respondents (*n* = 334).

Variables	Adjusted Odds Ratio (aOR)	[95% Confidence Interval]		*p*-Value
**Parental Age (Reference: Under 35 years)**				
35–38 years	1.20	0.49	2.90	0.692
38–43 years	1.77	0.73	4.31	0.206
Over 43 years	1.07	0.44	2.56	0.888
**Parental Education (Reference: No college degree)**				
College degree	2.12	1.04	4.36	0.039 *
**Household Income (Reference: Under USD 150,000/year**				
Over USD 150,000/year	0.70	0.37	1.30	0.260
**Practice Location (Reference: Philadelphia)**				
Suburban	0.38	0.21	0.67	0.001 *
**Child Age at Survey 1 (Reference: 16+ years)**				
12–15 years	1.06	0.39	2.76	0.915
5–11 years	0.55	0.20	1.39	0.216
Under 5 Years	0.32	0.09	1.04	0.061
**Child Influenza Vaccine Receipt 2021 (Reference: No)**				
Yes	4.07	2.08	8.12	<0.001 *
**COVID-19 Vaccine Safety Perception for Youngest Child (Reference: Not very safe)**				
Very safe	2.69	1.47	4.99	0.001 *
**Child Vaccination Location Preference (Reference: Other or No Preference)**				
At pediatrician/physician’s office	1.90	1.09	3.34	0.024 *

* Denotes significant *p*-value (α = 0.05).

## Data Availability

Not applicable.

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
