# Peer review of "Comparing COVID-19 Vaccination Outcomes with Parental Values, Beliefs, Attitudes, and Hesitancy Status, 2021–2022"

_vaccines, 2022, doi:10.3390/vaccines10101632_

Round 1
Reviewer 1 Report
Dear authors, the manuscript is moderately interesting and needs major revisions, to follow some considerations:
- please enter MeshTerms as Keywords and check if others can be added, in my opinion yes;
- the introduction is adequate and correctly shows the background;
- please explain better in materials and methods how the "CHOP Outpatient Care Network" is organized, for an international reader this could be useful to better understand the study;
- in materials and methods please indicate the regressions for which covariates have been adjusted and indicate the confidence interval used;
- the figures are clear and easy to read;
- it is not clear what the odds ratios are adjusted for, please enter in table 2 and in the text.
- the conclusions are trivial, the authors should better explain how this study should help the policy-maker to improve the situation;
- in table 1 it would be appropriate to calculate the Pearson correlation coefficient for the "parental characeteristics";
- for the logistic regression it would be advisable to have the "R-squared" (coefficient of determination);
- the biggest problem remains the sampling which is not optimal and it would be advisable for the authors to clarify how they thought to avoid possible sources of error, the response rate is really very low.
Please answer point by point.
kind regards
Author Response
Thanks for your helpful comments! See responses here:
- please enter MeshTerms as Keywords and check if others can be added, in my opinion yes;]
- Thanks, we have added the MeSH terms “COVID-19 Vaccines” and “Vaccination Hesitancy”
- please explain better in materials and methods how the "CHOP Outpatient Care Network" is organized, for an international reader this could be useful to better understand the study;
- We appreciate this comment. We are unsure what additional information would be relevant since the Methods section currently contains this text in lines 82-87: “The network included 31 pediatric primary care sites located within the greater Philadelphia regions: three academic and three non-academic sites located within Philadelphia and 25 suburban sites located across the Philadelphia (n=22) and New Jersey (n=3) suburbs (Supplemental Table 1). Academic sites have a higher proportion of uninsured and Medicaid patients and include physicians-in-training [23].” Supplemental Table 1 contains the demographic characteristics of the CHOP primary care network. We are happy to provide more information if necessary, but we think the current description will suffice.
- in materials and methods please indicate the regressions for which covariates have been adjusted and indicate the confidence interval used;
- The Methods sections explains this in lines 149-157: We modeled differences between parents whose age-eligible eldest child had versus had not received both doses of COVID-19 vaccine using a multivariable logistic regression model (α = 0.05) and have reported adjusted odds ratios with 95% confidence intervals and p-values. Variable selection for our final logistic regression was performed using Lasso regression (α = 1) [28]. All selected variables were checked for missingness (using a 5% maximum missing threshold) and collinearity (using a variance inflation factor threshold of less than 10). We performed sensitivity analyses by additionally running separate models for children ages 5-11 years and 12-15 years at the date of Survey 1 completion, based on the age groups approved by the FDA (Figure 1).
- it is not clear what the odds ratios are adjusted for, please enter in table 2 and in the text.
- By definition, an adjusted odds ratio is an odds ratio that controls for other predictor variables in a model. Therefore, given that we have multiple predictor variables in our logistic regression, these odds ratios are all adjusted odds ratios.
- the conclusions are trivial, the authors should better explain how this study should help the policy-maker to improve the situation;
- We appreciate the comment. Please see lines 274-280, 298-305, 317-320, 346-350, and 382-386.
- in table 1 it would be appropriate to calculate the Pearson correlation coefficient for the "parental characeteristics";
- Thanks for this suggestion, these p-values have been added using the appropriate statistical tests.
- for the logistic regression it would be advisable to have the "R-squared" (coefficient of determination);
- In our study, we have used a logistic regression (given our categorical outcome variable), and an equivalent statistic to R-squared does not exist. Only linear regressions have R-squared values.
- the biggest problem remains the sampling which is not optimal and it would be advisable for the authors to clarify how they thought to avoid possible sources of error, the response rate is really very low.
- We appreciate this important comment. We have expanded our Limitations section extensively to address the low response rate some more. See lines 354-381.
Reviewer 2 Report
Estimated Dr. SRIVASTAVA,
I've read with great interest this longitudinal study about the comparison of COVID-19 vaccination outcomes with the parental values, beliefs, attitudes, and hesitancy status. Briefly, Authors identified higher odds of accepting two doses of COVID-19 vaccine for their child was in respondents who had greater trust in COVID-19 vaccine efficacy (aOR [CI]: 2.69[1.47-4.99], p=0.001), or had previously vaccinated their child against influenza (aOR [CI]: 4.07[2.08-8.12], p<0.001). On the contrary, the odds of vaccinating their child were lower for respondents who attended suburban vs. urban practices (aOR [CI]: 0.38[0.21-0.67], p=0.001).
Even though these results were not unexpected as highly consistent with previous studies, Authors were able to perform a very well designed and reported study, and accurately addressed most of potential issues in the "limitation" section.
From the point of view of the present reviewer, only a minor amendments would be required, that is discussing in further details: 1) whether the sample could be considered representative or not when compared to the minimum sample size calculated by a proper power analysis; 2) how the reported data may have been affected by the very high rate of non-respondents (some glimpses are in fact discussed, i.e. selection bias, but a more extensive debate is required).
Author Response
Thanks for your helpful comments! See responses here:
1) whether the sample could be considered representative or not when compared to the minimum sample size calculated by a proper power analysis;
- Thanks for this comment. Since this was 1) a convenience sample of CHOP parents; 2) surveys had to be administered in a limited timeframe given FDA authorizations; and 3) we were not testing the differences in effect size based on an intervention, we did not perform power calculations. This limits the generalizability in terms of quantifying exact associations and causal relationships, but it can still quickly identify areas for improvement for providers. We have expanded the Limitations section to discuss this.
2) how the reported data may have been affected by the very high rate of non-respondents (some glimpses are in fact discussed, i.e. selection bias, but a more extensive debate is required).
- We appreciate this important comment. We have expanded our Limitations section extensively to address the low response rate some more. See lines 354-381.
Round 2
Reviewer 1 Report
The authors have modified according to the indications, as regards the R-square I meant the calculation of the Pseudo-R2 for logistic regressions, I'm sorry for the inconvenience.
Kind regards.